## RESEARCH ARTICLE

# Lamin B loss in nuclear blebs is rupture dependent whereas increased DNA damage is rupture independent

Catherine G. Chu[1,*], Nick Lang[1,*], Erin Walsh[1], Mindy D. Zheng[1], Gianna Manning[1], Kiruba Shalin[1], Lyssa M. Cunha[1], Kate E. Faucon[1], Nicholas Kam[1], Sara N. Folan[1], Arav P. Desai[1], Emily Naughton[1], Jaylynn Abreu[1], Alexis M. Carson[1], Zachary L. Wald[1], Dasha Khvorova-Wolfson[1], Leena Phan[1], Hannah Lee[1], Mai Pho[1], Kelsey Prince[1,2], Katherine Dorfman[1], Michael Seifu Bahiru[1,3,‡] and Andrew D. Stephens[1,2,‡]

## ABSTRACT

The nucleus must maintain shape and integrity to protect the function of the genome. Nuclear blebs are deformations identified by decreased DNA density that commonly lead to rupture. Lamin B levels often vary drastically between blebs. We tracked rupture via time-lapse imaging of nuclear localization sequence (NLS)–GFP immediately followed by immunofluorescence imaging of lamins and known rupture markers. We find that lamin B1 loss consistently marks ruptured nuclear blebs better than lamin A/C, emerin and cGAS. Visualizing post-rupture lamin B1 loss and emerin enrichment reveals that cell lines display widely different propensities for nuclear bleb rupture. To determine how rupture affects DNA damage, we time-lapse-imaged ruptured and unruptured blebs, then conducted immunofluorescence on the same cells for DNA damage markers γH2AX and 53BP1. We find that DNA damage is increased in blebbed nuclei independently of rupture. This was verified in blebbed LNCaP nuclei, which do not rupture and maintain lamin B1, but still show increased DNA damage. Thus, we confirm that lamin B is the most consistent marker of nuclear rupture, and that blebbed nuclei have increased DNA damage regardless of rupture.

KEY WORDS: Nucleus, Nuclear bleb, Nuclear rupture, Lamin B1, Emerin, DNA damage, cGAS

## INTRODUCTION

Since the advent of the microscope, nuclear morphology has been used to both diagnose and prognose human diseases including cervical, breast and prostate cancer. Nuclear blebs are a class of abnormal nuclear morphology associated with disease. In prostate cancer, the frequency of nuclear blebbing correlates with higher Gleason scores, a standard measure of disease severity (Helfand et al., 2012). Nuclear blebs arise from perturbations to the two major mechanical components of the nucleus – chromatin and lamins (Deviri et al., 2017; Hobson and Stephens, 2020; Kalukula et al.,

2022; Stephens et al., 2018a). When antagonistic forces overcome nuclear strength, it leads to nuclear shape deformations that cause high curvature and nuclear rupture (Chen et al., 2020; Denais et al., 2016; Hatch and Hetzer, 2016; Le Berre et al., 2012; Mistriotis et al., 2019; Stephens et al., 2018b; Xia et al., 2018; Zhang et al., 2019). Nuclear rupture has been reported to cause dysfunction via increased DNA damage, transcriptional changes and loss of cell cycle control (Kalukula et al., 2022). Despite these observations, the precise causal relationships between nuclear blebbing, rupture and their cellular consequences are yet to be fully elucidated.

The molecular composition of nuclear blebs in relation to nuclear rupture remains unclear. Previous work identified that DNA density loss is the most consistent marker of a nuclear bleb relative to the nuclear body across perturbations and different cell types (Bunner et al., 2025; Pujadas Liwag et al., 2025). In contrast, lamin B levels vary widely between cell types and perturbations (Jung-Garcia et al., 2023; Nmezi et al., 2019; Stephens et al., 2018b). The mechanisms driving this heterogeneity in lamin B distribution – where some blebs maintain normal levels while others show dramatic reductions – remain unclear.

Lamin B depletion in nuclear blebs could occur during nuclear rupture or during stretching of the lamina as the bleb forms. It is known that lamin B does not localize to nuclear ruptures induced by laser ablation (Kono et al., 2022; Sears and Roux, 2022). However, this has yet to be shown definitively in naturally occurring nuclear bleb-based ruptures. Alternatively, the growing nuclear bleb could cause loss of lamin B through dilution (Funkhouser et al., 2013; Pfeifer et al., 2022). Using time-lapse imaging immediately followed by immunofluorescence imaging (hereafter called 'time-lapse imaging into immunofluorescence'), we previously demonstrated that lamin B levels were significantly different in ruptured versus unruptured blebs (Bunner et al., 2025). Further validation and investigation are required to determine the ability of lamin B levels to be a marker of nuclear bleb rupture.

Nuclear rupture can be monitored by both dynamic visualization of nuclear localization sequence (NLS)–GFP during live-cell imaging and static population analysis using the enrichment of rupture-specific nuclear envelope proteins. NLS–GFP is concentrated in the nucleus and will spill into the cytoplasm upon nuclear rupture, but is re-enriched after nuclear envelope healing, which occurs in ~10 min (Le Berre et al., 2012; Pho et al., 2024a; Vargas et al., 2012; Young et al., 2020). Following nuclear envelope rupture, BAF (also known as BANF1) rapidly accumulates at the rupture site and recruits emerin, LEMD2, lamin A/C and ESCRT-III to repair the nuclear envelope (Halfmann et al., 2019; Janssen et al., 2022; Kono et al., 2022; Sears and Roux, 2022; Young et al., 2020). Recent studies have highlighted emerin as a reliable marker for detecting recent nuclear rupture events (Halfmann et al., 2019; Young et al., 2020).

[1]Biology Department, University of Massachusetts Amherst, Amherst, MA 01003, USA. [2]Molecular and Cellular Biology, University of Massachusetts Amherst, Amherst, MA 01003, USA. [3]Program in Neuroscience and Behavior, University of Massachusetts, Amherst, MA 01003, USA.
*These authors contributed equally to this work

‡Authors for correspondence (andrew.stephens@umass.edu; mbahiru@umass.edu)

A.D.S., 0000-0001-5474-7845

An alternative marker of nuclear rupture is cyclic GMP-AMP synthase (cGAS), a cytosolic double-stranded (ds)DNA sensor (Kono et al., 2022; Kovacs et al., 2023). Thus, nuclear rupture can be analyzed using a few different approaches.

Nuclear rupture has been shown to cause cellular dysfunction most consistently via increased DNA damage as measured by γH2AX or 53BP1 (also known as TP53BP1) foci (Chen et al., 2018; Denais et al., 2016; Irianto et al., 2016; Pho et al., 2024a; Raab et al., 2016; Shah et al., 2021; Stephens et al., 2019b; Xia et al., 2018). However, recent studies have demonstrated that nuclear deformation alone, even without rupture, induces DNA damage (Shah et al., 2021). This raises a crucial question about the relative contributions of nuclear blebbing and rupture to DNA damage accumulation, necessitating quantitative comparisons between ruptured and non-ruptured nuclear blebs.

We used novel time-lapse imaging into immunofluorescence to first track nuclear bleb rupture history, then assay the composition of the nuclear bleb relative to the nuclear body. We provide quantitative measurement of fidelity of nuclear rupture markers. Next, we used those nuclear bleb post-rupture markers to assay across a diverse array of mammalian cell lines (mouse and human). Finally, we tracked DNA damage in blebbed nuclei that did or did not rupture. Our novel findings provide a framework for others to quickly and efficiently determine the association between blebbing and increased DNA damage and to determine cell type or individual rupture history via lamin B levels in the nuclear bleb.

## RESULTS

### Lamin B loss in a nuclear bleb is the best marker for nuclear rupture

Nuclear blebs are >1 μm protrusions of the nucleus that exhibit decreased DNA density (Bunner et al., 2025) and commonly lead to nuclear rupture (Denais et al., 2016; Hatch and Hetzer, 2016; Stephens et al., 2019b; Vargas et al., 2012). Our previous work has revealed that lamin B1 is maintained in unruptured nuclear blebs, whereas nuclear blebs that have ruptured show loss of lamin B relative to the nuclear body. The fidelity of using loss of lamin B1 as a static marker for past rupture remains untested. We performed time-lapse imaging of mouse embryonic fibroblasts (MEFs) with NLS–GFP for 3 h at 2-min intervals to track rupture of blebbed nuclei. We then fixed those same cells and conducted immunofluorescence aided by a gridded dish (Fig. 1A). Ruptured nuclear blebs were assayed for levels of lamin A/C and lamin B1, and separately for the known nuclear rupture markers emerin and cGAS. For each nuclear bleb, we generated a nuclear bleb-to-body ratio to show protein enrichment (>1.3), maintenance (0.7–1.3) or loss (<0.7, Fig. 1B).

Lamin A/C staining of post-rupture nuclear blebs displayed heterogenous levels with loss occurring 60±8% (mean±s.e.m.) of the time (Fig. 1B,C). Analysis revealed that lamin levels were dependent on time elapsed since nuclear rupture. Nuclear blebs that ruptured <1 h before fixation showed 66% lamin A/C loss whereas nuclear blebs that ruptured >1 h before fixation showed only 36% loss (Fig. S1A,B). This agrees with multiple studies showing that lamin A/C is lost after nuclear rupture but re-recruited over time (Denais et al., 2016; Halfmann et al., 2019; Janssen et al., 2022; Kono et al., 2022; Sears and Roux, 2022; Shimi et al., 2008). However, other studies find enrichment of lamin C post-rupture (Kono et al., 2022). Thus, lamin A/C is an unreliable static marker for a post-rupture nuclear bleb given its heterogenous levels and time-dependent post-rupture recruitment.

In all imaged nuclei where a nuclear bleb ruptured, loss of lamin B1 occurred 100% of the time (Fig. 1B,C). To determine whether lamin B1 was present in unruptured nuclear blebs, we tracked events

for newly formed blebs that did not rupture. Nuclear blebs that formed and did not rupture during the time lapse maintained lamin B1 levels (Fig. 1D; Fig. S1C), recapitulating our previous findings (Bunner et al., 2025). Thus, lamin B1 loss is a highly reliable marker for past nuclear bleb rupture.

To compare lamin B1 loss as a marker for nuclear rupture, we turned to two known rupture markers – emerin and cGAS (Halfmann et al., 2019; Young et al., 2020). The nuclear envelope protein emerin and cytosolic DNA sensor cGAS use two different mechanisms to enrich nuclear blebs post-rupture. The emerin and cGAS nuclear bleb-to-body ratio showed enrichment (>1.3) in 83±4% and 74±7% of the time for post rupture nuclear blebs, respectively (Fig. 1B,C). While both maintain an average enrichment over time from rupture, cGAS enrichment decreased over time while emerin did not (Fig. S1D,E). Thus, even the most well-documented nuclear rupture markers did not match the 100% fidelity of lamin B1 loss upon nuclear rupture.

### Nuclear blebs consistently exhibit decreased DNA density while levels of rupture markers lamin B1 and emerin sort cell types into two categories

Nuclear blebs can be identified in static images through systematic scanning and DNA density measurements (Fig. 2A,B), as previously reported (Bunner et al., 2025). We conducted immunofluorescence on numerous cell types and perturbations to track whether these nuclear blebs had ruptured by assaying for nuclear rupture markers lamin B1 and emerin.

MEFs provide a cell line that displays nuclear blebs with a high rate of rupture, resulting in a majority of blebs displaying decreased levels of lamin B relative to the nuclear body (Fig. 2A,C). MEF cell treatments and genetic modifications provide the ability to compare wild-type (WT) cells to cells with chromatin perturbations or lamin perturbations that increase nuclear blebbing (Fig. S2D). Treatment of MEF cells with the histone deacetylase inhibitor VPA or histone methyltransferase inhibitor DZNep causes chromatin decompaction via increased euchromatin or decreased heterochromatin respectively, leading to increased nuclear blebbing and rupture (Kalinin et al., 2021; Stephens et al., 2018b, 2019b). Genetic knockout of lamin B1 and knockdown of lamin A in MEFs also increases nuclear blebbing and rupture (Berg et al., 2023; Hatch and Hetzer, 2016; Pho et al., 2024b; Vahabikashi et al., 2022; Vargas et al., 2012). Immunofluorescence of DNA, lamin B1 (or lamin B2), and emerin revealed that MEF nuclear blebs were heterogeneously decreased for lamin B1 whereas emerin levels could be enriched, maintained or depleted relative to the nuclear body (Fig. 2A,C,D). Again, DNA density via Hoechst 33342 staining remained consistently decreased in blebs across cells and perturbations (Fig. 2B), in agreement with our previous work (Bunner et al., 2025). Overall, WT MEFs and MEFs with perturbations show a majority lamin B1 loss and simultaneous emerin enrichment, signaling that most nuclear blebs rupture.

Human cancer cell types fall on a spectrum between loss and maintenance of lamin B1 in the nuclear bleb relative to the body (Fig. 2C). Immunofluorescence data revealed that human HT1080 fibrosarcoma and PC3 prostate cancer cell lines show a MEF-like heterogenous decrease of lamin B1 in the nuclear bleb relative to the nuclear body, with emerin levels that could be relatively enriched, maintained or depleted (Fig. 2C,D). By contrast, human cancer cell lines LNCaP, DU145 and HeLa cells retained similar levels of lamin B1 and emerin in the nuclear bleb compared to the body, suggesting that no rupture had occurred in those blebs (Fig. 2C,D). Thus, depending on the human cell type, nuclear bleb rupture markers show variable results.

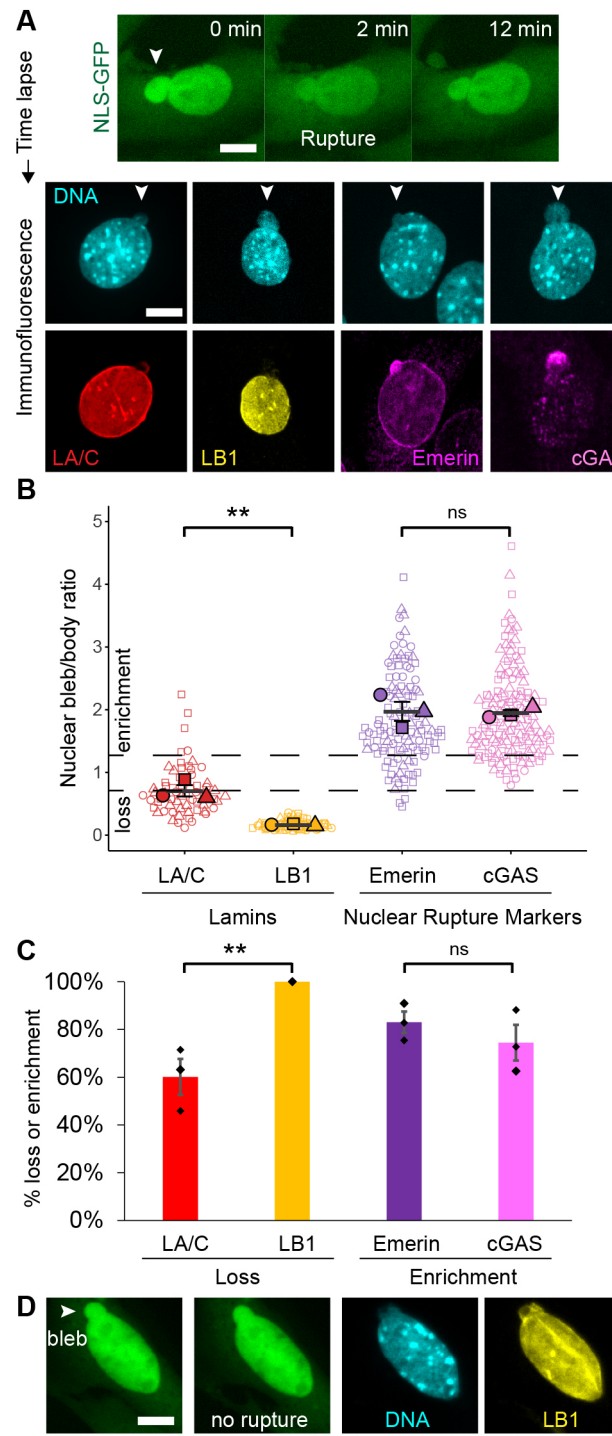

**Fig. 1. Time-lapse of nuclear rupture into immunofluorescence to determine post-rupture marker fidelity.** (A) Example images of MEF time lapse NLS–GFP imaging to track nuclear rupture into immunofluorescence via Hoechst 33342 DNA stain (cyan) and lamin A/C (red), lamin B1 (yellow), emerin (magenta) or cGAS (pink). Arrowheads denote the nuclear bleb. (B) Super plots of post-rupture nuclear bleb relative intensity nuclear bleb-to-body ratio (bleb/body) for lamin A/C, lamin B1, emerin and cGAS. Dotted lines denote loss (<0.7) and enrichment (>1.3) thresholds for bleb-to-body ratio. (C) Each biological replicate from B was turned into a percentage for loss of lamin or enrichment of emerin or cGAS to mark a nuclear bleb that ruptured. Each condition was assayed in triplicate with lamin A/C and lamin B1 $n$=19, 24, 21; emerin $n$=22, 69, 29; and cGAS $n$=15, 55, 76. (D) Time-lapse into immunofluorescence for a nuclear bleb that was newly formed and did not rupture shows maintenance of lamin B1. Tracking of 10 nuclei that did not rupture yielded an average lamin B1 nuclear bleb-to-body ratio of 0.93±0.07 (mean±s.e.m.; Fig. S1C). Mean±s.e.m. is graphed in B and C. \*\*$P$<0.01; ns, no significance $P$>0.05 (two-tailed unpaired Student's $t$-test). Scale bars: 10 μm.

## Lamin B and emerin levels are correlated in the nuclear bleb

We hypothesized that nuclear rupture events should lead to a loss of lamin B1 and accumulation of emerin levels within nuclear blebs. This hypothesis is supported by our observation that ruptured nuclear blebs have decreased lamin B1 and increased emerin levels (Fig. 1). To test this, we measured the correlation of lamin B1 decrease with emerin enrichment that occurs at sites of nuclear rupture (Halfmann et al., 2019; Young et al., 2020). Across MEF WT and MEFs in perturbed conditions, nuclear blebs showed a heterogenous decrease of lamin B1 relative to the body (Fig. 3A). The nuclear bleb composition for each condition was graphed as the lamin B1 versus emerin bleb-to-body ratio (Fig. 3C–L). In 90% of nuclear blebs that maintained lamin B1 levels, emerin levels were also maintained (Fig. 3C–F). In contrast, blebs with decreased lamin B1 frequently showed substantial emerin enrichment, often several-fold higher than the nuclear body. However, not all nuclear blebs with decreased lamin B1 showed emerin enrichment, as some showed similar or reduced emerin in the nuclear bleb relative to the body (Fig. 3M). Overall, we find that rupture marker emerin does not enrich in nuclear blebs that maintain lamin B1, whereas blebs with decreased lamin B1 are frequently enriched in emerin.

We next investigated human cell lines presenting heterogeneously decreased lamin B levels in the nuclear bleb to determine whether this phenomenon is reproducible in other cell types. The human cancer cell line HT1080 is well reported to have nuclear blebs that are rupture prone (Stephens et al., 2019b). Upon analysis of HT1080 nuclear blebs, we found a relationship between lamin B1 and emerin similar to that observed in MEFs (Fig. 3B). For blebs that had a similar level of lamin B1 in the bleb and body (bleb-to-body ratio=1), emerin levels in the nuclear bleb and body were also similar (Fig. 3H). Decreased lamin B1 in nuclear blebs of HT1080 nuclei were associated with both emerin enrichment and reduction. This trend was also present in prostate cancer cell line PC3 (Fig. 3I) and previously reported in human fibroblast cells (Janssen et al., 2022). These findings recapitulate that bleb-based rupture-prone nuclei display loss of lamin B1 in conjunction with enrichment of emerin due to nuclear rupture.

Human cell lines LNCaP, DU145 and HeLa provide the opportunity to investigate nuclear blebs that maintain lamin B1 relative to the body (Figs 2 and 3J–L). Correlation graphs of lamin B1 and emerin levels in LNCaP, DU145 and HeLa cells showed that there was rare enrichment of emerin, in agreement with lamin B1 maintenance. Interestingly, the relationship between lamin B1 and emerin levels in the bleb followed a linear correlation, suggesting a general thinning of the nuclear lamina and envelope. These levels were also centered around 1 relative to the rest of the nucleus. Thus,

Overall, we find two nuclear bleb rupture capacity categories across cell types (a versus b, Fig. 2C; Fig. S2). MEFs, HT1080 and PC3 display a category of nuclear blebs with decreased levels of lamin B and increased emerin in the nuclear bleb, indicative of nuclear rupture. By contrast, LNCaP, DU145, and HeLa displayed a different category of nuclear blebs that maintain lamin B and emerin levels, indicative of a lack of nuclear rupture. A portion of these groups also show differential emerin behavior with enriched or maintained levels respectively (a versus b, Fig. 2D). These data suggest that nuclear blebs can have widely different capacities for rupture based on cell type.

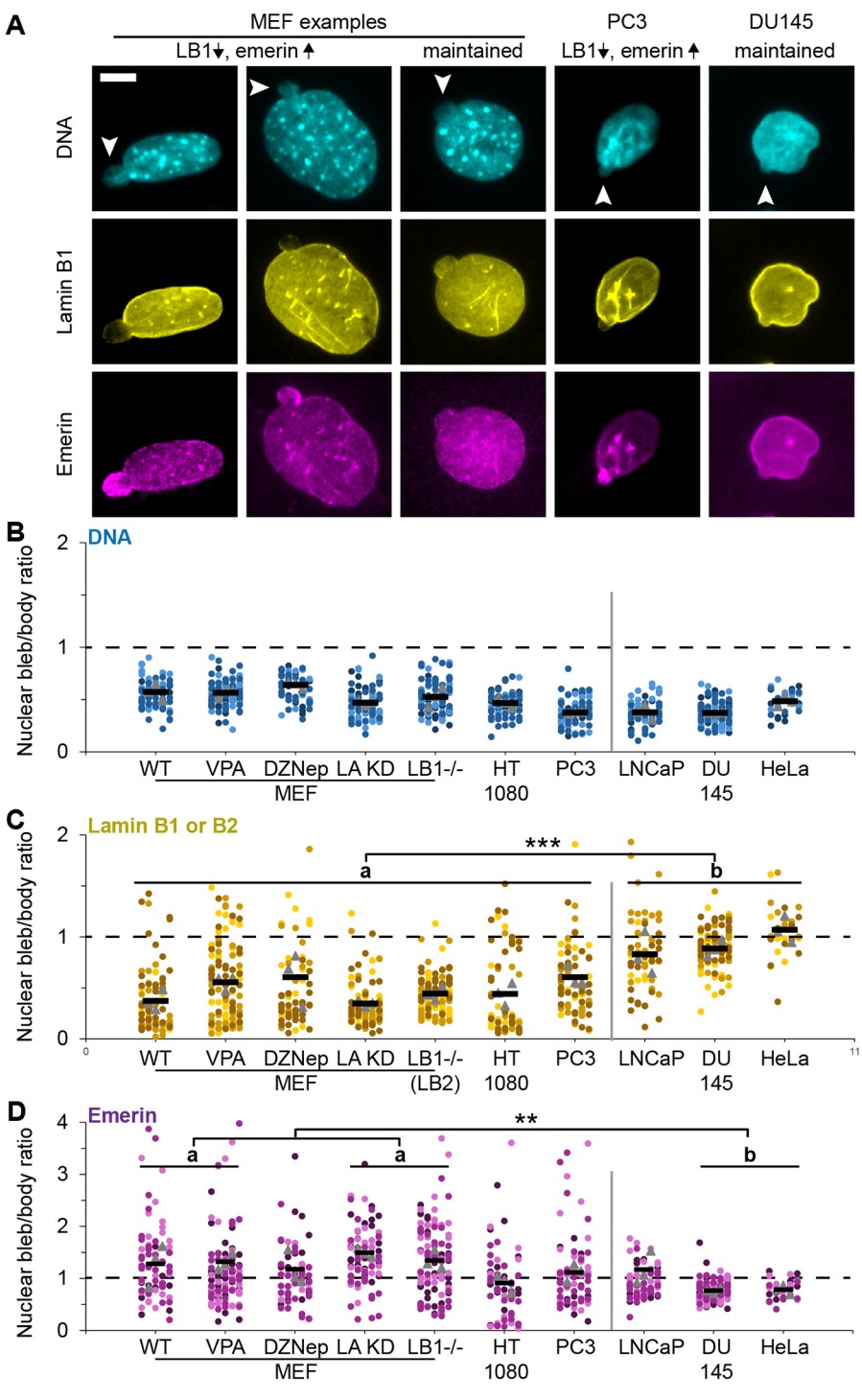

**Fig. 2. Nuclear bleb versus body levels reveal consistent loss of DNA while nuclear rupture markers lamin B and emerin levels vary across cell types.** (A) Example images of a blebbed nucleus in MEF, PC3 and DU145 cells labeled for DNA via Hoechst 33342 (cyan) and immunofluorescence of lamin B1 (yellow) and emerin (magenta). Arrowheads denote the nuclear bleb. (B–D) Super plots of the nuclear bleb-to-body ratio (bleb/body) relative intensity ratios for (B) DNA, (C) lamin B and (D) emerin in MEF WT, and upon chromatin decompaction (VPA and DZNep), and lamin perturbations [LA KD and LB1$^{-/-}$ (Lmnb1$^{-/-}$)] and in human cell lines HT1080, PC3, LNCaP, Du145 and HeLa. Each condition was assayed in triplicate with $n>10$. **$P<0.01$, ***$P<0.001$ (no asterisk denotes no significance; one way ANOVA with Tukey's post-hoc test). Tables of all comparisons and blebbing percentages in Fig. S2. Asterisks in C and D indicate that similar groups a and b are statistically different for all in the group. $N=3–4$, where MEF WT, $n=11, 16, 16, 34$; VPA $n=38, 35, 36$; DZNep $n=19, 10, 32$; LA KD $n=29, 29, 27$; Lmnb1$^{-/-}$ $n=33, 35, 41$; HT1080 $n=18, 15, 29$; PC3 $n=16, 29, 32$; LNCaP $n=13, 27, 27$; DU145 $n=37, 40, 35$; HeLa $n=11, 10, 10$. Mean±s.e.m. is graphed in B–D. Scale bar: 10 µm.

nuclear blebs that maintain normal lamin B1 levels consistently show unenriched emerin distribution, indicating an absence of nuclear rupture events.

### Time-lapse imaging into immunofluorescence reveals that increased DNA damage in nuclear blebs is rupture independent

Previous studies of immunofluorescence report that blebbed nuclei have higher levels of DNA damage than normally shaped nuclei (Pho et al., 2024a; Stephens et al., 2019b). Numerous publications hypothesize that blebbed nuclei are associated with increased DNA

damage due to nuclear rupture (Kalukula et al., 2022; Stephens, 2020). To test this hypothesis, we compared DNA damage levels across normally shaped nuclei (non-blebbed and unruptured), blebbed nuclei that ruptured and blebbed nuclei that did not rupture during the time lapse. We imaged MEF NLS–GFP cells at 2-min intervals for 3 h to track nuclear rupture, then immediately conducted immunofluorescence of DNA damage markers γH2AX and 53BP1 (Fig. 4A).

Normally shaped nuclei maintained a low level of DNA damage marker γH2AX relative to blebbed nuclei (Fig. 4A–C), recapitulating our previous static studies (Pho et al., 2024a;

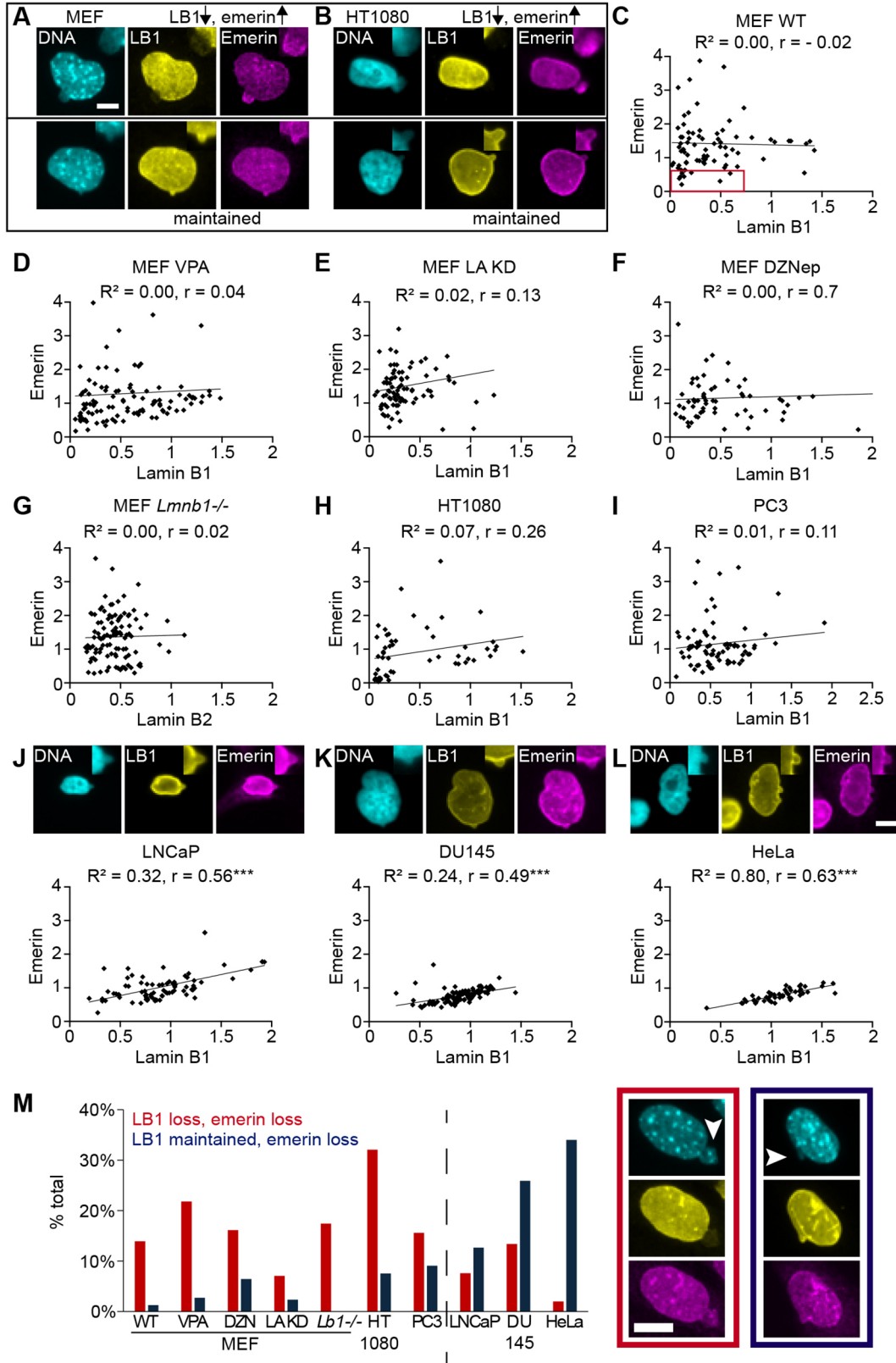

**Fig. 3.** See next page for legend.

Stephens et al., 2019b). Unexpectedly, the blebbed nuclei showed similarly elevated DNA damage levels regardless of rupture status (Fig. 4B,C). We found no correlation between γH2AX DNA damage levels and number of nuclear ruptures, the time elapsed since the last nuclear rupture or emerin enrichment levels (Fig. S3A, B). DNA damage foci were found to randomly distribute in the nucleus and were not consistently located in the nuclear bleb. A second cell type, human cancer cell line HT1080, also showed that

**Fig. 3. Nuclear blebs with decreased lamin B1 display increased emerin while maintenance of lamin B1 shows correlation with emerin levels.** (A,B) Example images of nuclear blebs with or without changes (arrows) in lamin B1 (LB1) and emerin for (A) MEF and (B) human HT1080 cell lines. DNA is stained with Hoechst 33342 (cyan). Top right insets show a closeup of the bleb. (C–L) Quantitative analysis of nuclear bleb-to-body (bleb/body) ratios comparing lamin B versus emerin levels across cell lines: MEF wild type (WT, $n$=79), MEF VPA ($n$=110), MEF LA KD ($n$=85), MEF DZNep ($n$=61), MEF $Lmnb1^{-/-}$ ($n$=109), HT1080 ($n$=53), PC3 ($n$=77), LNCaP ($n$=79), Du145 ($n$=112), HeLa ($n$=31). Cell lines (A) MEF and (B) HT1080 show nuclear blebs with decreased lamin B1 and increased emerin whereas cell lines (J) LNCaP, (K) Du145 and (L) HeLa that maintain lamin B1 levels in the bleb show a linear relation of emerin nuclear bleb-to-body ratio around 1. Each set of data was fit for linear regression denoted by $R^2$ value and statistically significant correlation via Pearson's correlation ($r$). ***$P$<0.001. (J–L) Example images along with graphs of emerin relative to lamin B1 of LNCaP, Du145, and HeLa nuclear blebs stained for DNA via Hoechst 33342 (cyan) and immunofluorescence of lamin B1 (yellow) and emerin (magenta) that largely maintain similar levels in the nuclear bleb and body and have a strong Pearson's correlation ($r$). Top right insets show a closeup of the bleb. (M) Graph of percentage of total nuclear blebs with decreased emerin nuclear bleb/body ratio along with lamin B1 decreased (red) or maintained (dark blue). Example images shown to the right; arrowheads highlight blebs. Red box in C denotes decreased lamin B1 and emerin. Scale bars: 10 µm.

elevated DNA damage levels were associated with blebbed nuclei independently of emerin enrichment in the blebs, further indicating no link between DNA damage and rupture (Fig. S3C). Finally, using CDT1 as a reporter of G1 and early S phase revealed that the increase in γH2AX occurs independently of the cell cycle (Fig. S3D). Taken together, this data supports that DNA damage association with nuclear blebs is independent of nuclear rupture.

To verify this finding, we conducted immunofluorescence using 53BP1, a second DNA damage marker. Time-lapse tracking of nuclear rupture via NLS–GFP into immunofluorescence of 53BP1 recapitulated a similar increase in DNA damage associated with blebbed nuclei, that again was independent of nuclear rupture (Fig. 4D–F). Thus, nuclear blebbing is associated with increased DNA damage, as shown using two different DNA damage markers but is independent of rupture.

### LNCaP blebbed nuclei lack ruptures but have increased DNA damage

LNCaP nuclei provide the possibility to explore DNA damage levels in nuclear blebs that rupture infrequently if at all. LNCaP nuclei have maintained lamin B levels in their nuclear blebs and show little to no emerin enrichment (Fig. 3J). We imaged LNCaP cells transiently transfected with NLS–GFP and found that no imaged nuclear blebs ruptured over 22.5 h of imaging (Fig. 5A, $n$=11 blebbed nuclei). Next, we assayed normal and blebbed nuclei for increased DNA damage. We found that blebbed LNCaP nuclei, which overall do not rupture, had increased levels of DNA damage relative to normally shaped nuclei (Fig. 5B,C). Thus, data from cell types that display nuclear bleb ruptures frequently and infrequently show that increased DNA damage association in blebbed nuclei is not dependent on nuclear rupture.

### DISCUSSION

Providing clear markers for nuclear blebs and nuclear bleb rupture aids the ongoing understanding of how and why nuclear blebs form and rupture. We again show that DNA density levels are the most consistent marker of a nuclear bleb (Fig. 2B) in strong agreement with our previously published work (Bunner et al., 2025; Pujadas Liwag et al., 2025). Furthermore, we clarify that lamin B1 presence or absence in the nuclear bleb is determined by nuclear rupture

history as evidenced by data collected using time-lapse imaging into immunofluorescence (Fig. 1).

Nuclear blebs are highly consequential features of cellular pathology, contributing to nuclear dysfunction that might promote disease progression (Kalukula et al., 2022; Stephens et al., 2019a). We provide novel data showing that the presence of a nuclear bleb denotes dysfunction that is independent of nuclear rupture (Figs 4 and 5). This is a paradigm shift from the prior assertion that nuclear rupture was the main cause of dysfunction. These results both elucidate why lamin B1 levels in the nuclear bleb are variable between cell types and reveal that the presence of a nuclear bleb alone is a sufficient marker of underlying nuclear dysfunction via DNA damage.

### Lamin B1 loss in nuclear blebs indicates prior rupture

Previous studies have grappled with both the significance and heterogeneity of lamin B1 loss in nuclear blebs. Our data clarifies that loss of lamin B1 is a marker for past nuclear bleb rupture via NLS–GFP time-lapse into immunofluorescence (Fig. 1). Furthermore, our data show that lamin B1 loss heterogeneity in the nuclear bleb is due to the fact that not all nuclear blebs rupture and that different cell types have differential capacities for rupture (Figs 2, 3 and 5). Although our data is definitive regarding the ability of lamin B1 loss to report past nuclear bleb rupture, the mechanism requires further consideration.

One hypothesis is that lamin B1 is lost upon nuclear rupture. Other studies have detailed the correlation between loss of lamin B1 and emerin enrichment in the nucleus, which is reliant on BAF recruitment of emerin post nuclear rupture (Janssen et al., 2022). This hypothesis is consistent with laser ablation studies where lamin B1 is permanently lost upon ablation of the nuclear envelope while lamin A and C are lost at rupture but are recruited back to the site of rupture (Kono et al., 2022; Sears and Roux, 2022). Naturally occurring ruptures have also been measured via loss of lamin B1 and cGAS (Kovacs et al., 2023). This selective loss likely results from unique integration of lamin B into the nuclear envelope via its farnesylation group, which is absent in lamins A and C (Butin-Israeli et al., 2012), as well as lack of a possible re-recruitment post-rupture. The loss of envelope-associated proteins following rupture aligns with previous observations (Hatch and Hetzer, 2016). These results support the idea that lamin B1 is lost upon rupture.

An alternative hypothesis is that during nuclear bleb formation, lamin B1 is diluted, which could cause the nuclear bleb to rupture (Funkhouser et al., 2013; Pfeifer et al., 2022). Indentation of the nucleus has been shown to dilute the lamina (Ivanovska et al., 2023). Our data supports that some nuclear blebs do show dilution of the lamina and nuclear envelope, as cell types that have blebs that rarely rupture can show a linear decrease of both lamin and emerin (Fig. 3). However, the presence of lamin B1 in unruptured nuclear blebs refutes this hypothesis (Fig. 1D; Fig. S1C). In this hypothesis, loss of lamin B should increase nuclear ruptures. Published data refute this, as loss of lamin B1 via knockout ($Lmnb1^{-/-}$) does not significantly increase nuclear bleb rupture frequency compared to other chromatin or lamin perturbations (Berg et al., 2023; Pho et al., 2024a; Stephens et al., 2018b).

### Nuclear blebs signal dysfunction independently of rupture

Previous studies proposed that nuclear rupture is the primary cause of cellular dysfunction in nuclei that are mechanically perturbed and display abnormal shape. Specifically, loss of nuclear integrity was thought to cause dysfunction through diffusion of nuclear proteins out of the nucleus and cytosolic proteins into the nucleus (Denais

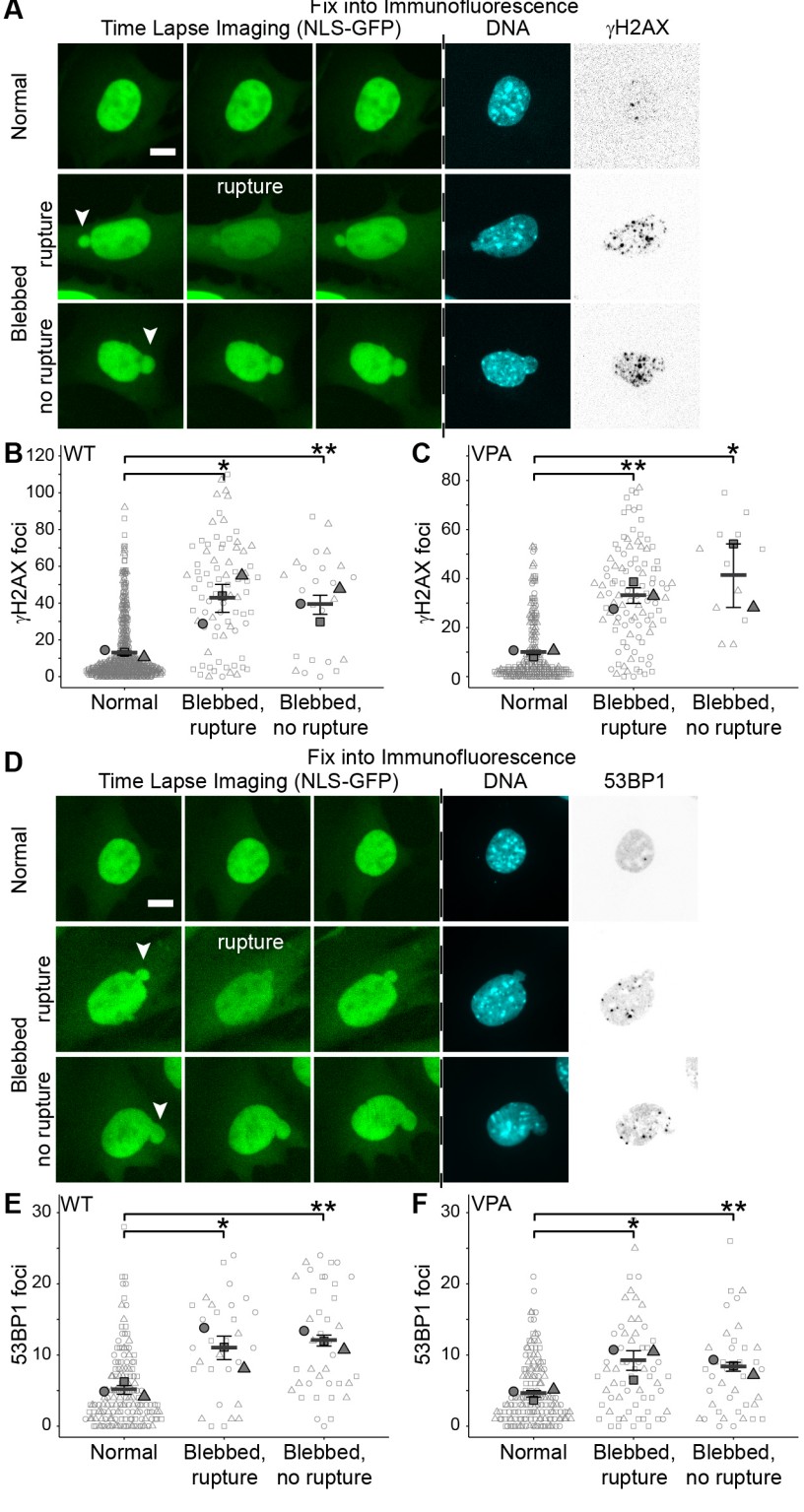

**Fig. 4. DNA damage increases in blebbed nuclei independently of nuclear rupture.** (A,D) Example images of MEFs from time-lapse imaging via NLS–GFP and subsequent immunofluorescence of the same nucleus. NLS–GFP images from time=0, 2 and 12 min. Immunofluorescence stained for DNA via Hoechst 33342 (cyan) and DNA damage foci labeled by (A) γH2AX or (D) 53BP1. Arrowheads highlight blebs. (B,C) Super plots of the number of DNA damage foci measured via γH2AX for (B) MEF WT or (C) MEF VPA-treated in nuclei that are normally shaped and unruptured, blebbed and rupture, and blebbed and no rupture (MEF WT normal *n*=186, 173, 112; bleb rupture *n*=9, 44, 26; bleb no rupture *n*=11, 5, 8; MEF VPA normal *n*=53, 120, 56; bleb rupture *n*=29, 42, 32; bleb no rupture *n*=7, 5). (E,F) Super plots of the number of DNA damage foci measured via 53BP1 for (E) MEF WT or (F) MEF VPA-treated in nuclei that are normally shaped and unruptured, blebbed that rupture, and blebbed with no rupture (MEF WT normal *n*=50, 50, 50; bleb rupture *n*=11, 10, 10; bleb no rupture *n*=10, 16, 12; MEF VPA normal *n*=50, 50, 50; bleb rupture *n*=15, 27, 21; bleb no rupture *n*=12, 14, 15). *P<0.05, **P<0.01 (no asterisk denotes no significance, P>0.05; two-tailed unpaired Student's *t*-test). Mean±s.e.m. is graphed (B,C,E,F). Scale bars: 10 µm.

et al., 2016; Nader et al., 2021; Raab et al., 2016; Xia et al., 2018). DNA damage emerged as the most frequently reported dysfunction associated with altered nuclear mechanics and morphology. Our previous work demonstrated both increased DNA damage in blebbed nuclei and frequent rupture of nuclear blebs (Pho et al., 2024a; Stephens et al., 2019b). However, our new data reveal that blebbed nuclei that do and do not rupture have a similar increase in DNA damage (Figs 4 and 5). This provides the novel finding that

blebbed nuclei are associated with increased DNA damage regardless of rupture status.

One hypothesis for DNA damage associated with blebbed nuclei is that the mechanical and/or structural deformation caused by a nuclear bleb causes DNA damage. Supporting evidence for deformation-driven DNA damage exists in earlier literature. In one of the first papers to report that nuclear rupture causes DNA damage, an example image shows that deformation results in

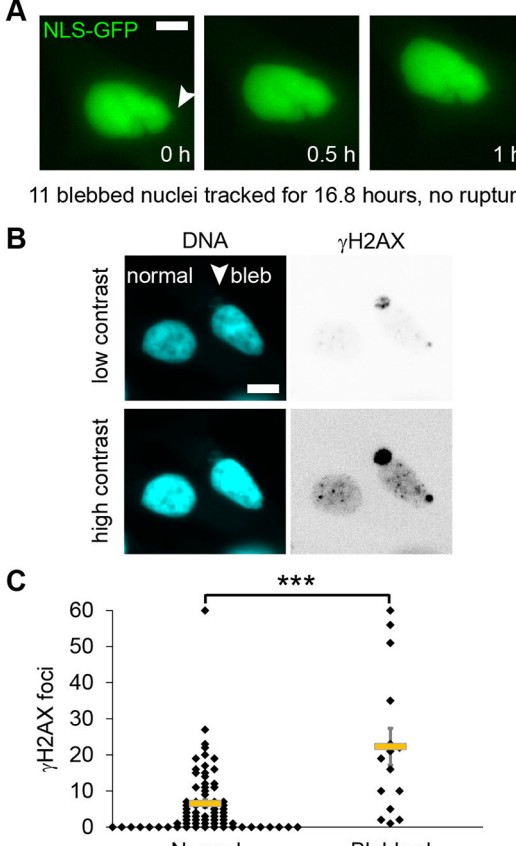

**Fig. 5. LNCaP blebbed nuclei do not show nuclear rupture but have increased DNA damage.** (A) Example images of a LNCaP blebbed nucleus (arrowhead highlights bleb) that does not show loss of NLS–GFP during live cell imaging (two biological replicates, total 11 blebbed nuclei imaged for 22.25 h showed 0 nuclear ruptures). (B) Example images of DNA labeled via Hoechst 33342 (cyan) and γH2AX DNA damage marker foci (inverted grayscale) shown in regular and high contrast to help visualize the bleb (arrowhead). (C) Graph of normal elliptical shaped and blebbed LNCaP nuclei DNA damage foci marked by γH2AX (normal $n$=75 and blebbed $n$=15). ***$P$<0.001 (two-tailed unpaired Student's $t$-test). Mean±s.e.m. is graphed. Scale bars: 10 μm.

increased DNA damage foci well before the nucleus ruptures (Denais et al., 2016). Increased DNA damage has also been reported in nuclei that are mechanically deformed but not ruptured, reportedly due to DNA replication stress (Shah et al., 2021) and mechanical strain (Nava et al., 2020). This finding is further supported by work showing that glioblastoma cell types U251MG and U87MG both have nuclear blebs, but that the more replication-active U251 cells show more DNA damage (Kamikawa et al., 2023). This observation aligns with other reports showing that DNA damage in blebbed and ruptured nuclei is due to DNA replication stress (Shah et al., 2021; Xia et al., 2019). A non-mutually exclusive mechanism could be that nuclear deformation separates repair factors from their needed site of function (Irianto et al., 2016). These findings collectively suggest that the combination of active DNA replication and mechanical stress in blebbed nuclei, rather than rupture alone, drives DNA damage accumulation.

An alternative hypothesis is that DNA damage drives nuclear blebbing. Recent studies have highlighted the entangled relationship of DNA damage, nuclear blebbing and nuclear rupture. DNA damage drugs have been shown to cause increased nuclear blebbing and/or rupture (Eskndir et al., 2025; Kovacs et al.,

2023). However, the effect stems from the DNA damage response and not the DNA damage directly. Increased nuclear blebbing is suppressed by inhibition of ATM kinase, which signals during DNA damage repair to decrease heterochromatin, thus softening nuclei to cause more blebbing (Eskndir et al., 2025). Nuclear ruptures can also be suppressed by inhibition of ATR, the other DNA damage signaling kinase, which is reported to modify lamins (Kovacs et al., 2023). In both of these reports, DNA damage levels remained high upon ATM or ATR inhibition, but nuclear blebbing and/or rupture was suppressed, suggesting that it is not the DNA damage but instead the signaling response that alters chromatin or lamin levels and mechanical protection of nuclear shape and integrity. Furthermore, DNA damage across the cell population does not significantly increase in response to many other treatments that result in increased nuclear blebbing (Berg et al., 2023; Hatch and Hetzer, 2016; Mistriotis et al., 2019; Pho et al., 2024a; Stephens et al., 2019b).

### Nuclear blebs as a site of genome instability

The nuclear bleb might be a site of increased genome instability that drives a more aggressive disease state. Nuclear blebs are diagnostic markers of disease progression in prostate cancer (Helfand et al., 2012). Our studies of LNCaP cells, an early-stage prostate cancer model, revealed nuclear blebs that rarely rupture (Figs 2, 3 and 5). However, blebbed LNCaP nuclei show increased DNA damage via markers in both the nuclear body and the bleb. Progression of prostate cancer has also been linked to DNA translocations in androgen-responsive genes. LNCaP cells treated with DHT increase nuclear blebbing and activation of androgen-responsive genes that disproportionately end up in the nuclear bleb (Helfand et al., 2012). Other work shows the importance of transcriptional activity in causing nuclear blebbing (Berg et al., 2023; Prince et al., 2025 preprint). Thus, we posit that increased transcriptional activity aids nuclear deformation. Thus, the nuclear bleb might provide a unique environment that exacerbates genome instability, independently of nuclear rupture.

Paradoxically, the absence of rupture might exacerbate cellular dysfunction. Specifically, the lack of rupture might stop the enrichment of BAF, emerin, LEMD2 and ESCRT-III. BAF and emerin have important chromatin interaction roles that, upon their local enrichment in a ruptured bleb, aid compaction and reorganization (Alfert et al., 2019; Marano and Holaska, 2022), and general loss of emerin can lead to nuclear invasiveness (Hansen et al., 2024). In conclusion, the absence of rupture could remove possible points during which the nuclear environment could be repaired through multiple different mechanisms.

Future work should aim to determine the underlying cause of dysfunction in both nuclei with blebs and the microenvironment within the nuclear bleb. Does the mechanical state of having a nucleus herniate to form a bleb suggest that the genome is under physical stress, causing dysfunction? Does the nuclear bleb have a different or separate microenvironment? Although one study has found that compression with atomic force microscopy can cause increased DNA damage, more and differing experiments are needed to assess this possibility (Shah et al., 2021). For example, micropipette aspiration (Irianto et al., 2016; Zhang et al., 2019), nuclear confinement (Le Berre et al., 2012; Nader et al., 2021) or experiments with single (Neelam et al., 2015) or dual micropipette micromanipulation (Currey et al., 2022) could apply different types and modify global versus local deformations. Although we have reported that the nuclear bleb has consistently decreased DNA density (Bunner et al., 2025) and destabilized packing domains

(Pujadas Liwag et al., 2025), future studies of chromatin dynamics, protein diffusion, and *in situ* chromosome tracking would greatly aid the determination of the nuclear bleb microenvironment.

## MATERIALS AND METHODS

### Cell culture

Mouse embryonic fibroblasts (MEFs) were previously described (Shimi et al., 2008; Stephens et al., 2018b; Vahabikashi et al., 2022). MEF wild-type (WT), lamin A knockdown (LA KD), and lamin B1 knockout (*Lmnb1$^{-/-}$*) cells were cultured in DMEM (Corning) completed with 10% fetal bovine serum (FBS; HyClone) and 1% penicillin/streptomycin (P/S; Corning), incubated at 37°C and 5% $CO_2$. Cells were passaged after reaching 80–90% confluency or every 2 to 3 days. Cells were treated with 0.25% Trypsin, 0.1% EDTA without sodium bicarbonate (Corning), replated, and diluted with DMEM. Human fibrosarcoma HT1080 cells and HeLa human cervical cancer cells were cultured and passaged similarly. HT1080 and HeLa cells were obtained from ATCC.

Three human prostate cancer cell lines were used: LNCaP, DU145 and PC3 (American Tissue Culture Collection ATCC). DU145 and LNCaP cells were cultured in RPMI 1640 (Cytiva) with 10% FBS and 1% P/S. PC3 cells were cultured in DMEM completed with 10% FBS and 1% P/S.

### Drug treatments

MEF WT cells were treated with either 4 mM valproic acid (VPA, 1069-66-5, Sigma; Gurvich et al., 2004) or 1 µM 3-deazaneplanocin (DZNep, Cal Biochem; Miranda et al., 2009) for 24 h before time-lapse imaging or fixation for immunofluorescence. HT1080 and HeLa cells were treated with VPA similarly to MEF cell lines.

### Immunofluorescence

Cells were grown on coverslips in preparation. Cells were fixed with 3.2% paraformaldehyde and 0.1% glutaraldehyde in phosphate-buffered saline (PBS) for 10 min. Between steps, cells were washed three times with PBS with Tween 20 (0.1%) and azide (0.2 g/l) (PBS-Tw-Az). Next, cells were permeabilized by 0.5% Triton-X 100 in PBS for 10 min. Again, cells were washed with PBS-Tw-Az. Humidity chambers were prepared using Petri dishes, filter paper, sterile distilled water and parafilm. 50 µl of primary antibody solution was applied to the parafilm and the cell-side of the coverslip was placed on top to incubate for 1 h at 37°C in the humidity chamber. Primary antibodies used were rabbit anti-Lamin B1 (1:1000, ab16048 Abcam), mouse anti-emerin (1:2000, NCL-emerin, Leica Biosystems) and cGAS rabbit mAb D3O8O (1:250, 31659s Cell Signaling Technology). The humid chambers were removed from the incubator and the coverslips were washed in PBS-Tw-Az. New humidity chambers were made for the secondary incubation. 50 µl of secondary antibody was placed on the parafilm, then the coverslips were placed cell-side down and incubated in humidity chambers at 37°C for 30 min. The secondary antibody solution contained goat anti-rabbit-IgG conjugated to Alexa Fluor 488 (1:200, 4412s, Cell Signaling Technology) and goat anti-mouse-IgG conjugated to Alexa Fluor 555 (1:200, 4409s, Cell Signaling Technology). Afterwards, coverslips were washed with PBS-Tw-Az and placed cell-side down on a slide with a drop of mounting medium containing DAPI. Alternatively, cells were stained with a 1 µg/ml (1:10,000) dilution of Hoechst 33342 (Life Technologies) in PBS for 5 min, washed with PBS three times, and mounted with ProLong Gold Antifade (Invitrogen). Slides were allowed to cure for four days at 4°C before imaging.

For live-cell imaging into immunofluorescence experiments, we used gridded imaging dishes (Cellvis #D35-14-1.5GI). Similarly, cells were fixed in the gridded imaging dish with 4% paraformaldehyde for 15 min. Next, cells were washed in PBS, permeabilized by 0.1% Triton-X 100 in PBS for 15 min, and neutralized by 0.06% Tween 20 for 5 min. Again, cells were washed with PBS. Cells were then blocked with 2% bovine serum albumin (BSA) for 1 h. 500 µl of primary antibody solution was applied to the cells at room temperature for 2 h. Primary antibodies used were rabbit anti-lamin B1 (1:1000, ab16048, Abcam), mouse anti-emerin (1:1000, NCL-emerin-A, Leica Biosystems), rabbit anti-53BP1 (1:100, 4937s Cell Signaling Technology), rabbit anti-cGAS mAb D3O8O (1:250, 31659s Cell

Signaling Technology) or mouse anti-lamin A/C (1:200, 4777s Cell Signaling Technology). The cells were again washed with PBS. 500 µl of secondary antibody was then applied to the cells at room temperature for 1 h. Secondary antibodies used were either Alexa Fluor 555 anti-mouse IgG (1:1000, 4410s, Cell Signaling Technology) or Alexa Fluor 647 anti-mouse IgG (1:1000, 4413s, Cell Signaling Technology). After the cells were washed with PBS, 500 µl of conjugate antibody was applied to the cells at room temperature for 2 h. The conjugate antibody used was γH2AX–Alexa Fluor 647 rabbit (1:1000, 9720, Cell Signaling Technology). Cells were then washed with PBS and stained with 1 µg/ml (1:10,000) dilution of Hoechst 33342 (H3570, Invitrogen) in PBS for 15 min. Cells were washed and stored in PBS or mounted using ProLong Gold Antifade (Invitrogen) until imaging. Transmitted light images were acquired to align the gridded dish after fixation so that the same cells from the time-lapse imaging could be imaged for their respective immunofluorescence of DNA via Hoechst 33342, emerin, cGAS, lamin B1, lamin A/C, yH2AX and 53BP1.

### Immunofluorescence imaging

Immunofluorescence images were acquired using a QICAM Fast 1394 Cooled Digital Camera, 12-bit, Monochrome CCD camera (4.65×4.65 µm pixel size and 1.4 MP, 1392×1040 pixels) using Micromanager and a 40× objective lens on a Nikon TE2000 inverted widefield fluorescence microscope. Cells were imaged using transmitted light to find the optimal focus on the field of view to observe the nuclear blebs. Ultraviolet light (excitation 360 nm) was used to visualize DNA via DAPI, blue fluorescent light (excitation 480 nm) was used to visualize lamin B1, and green fluorescent light (excitation 560 nm) was used to visualize emerin. Green fluorescent light (excitation 560 nm) was also used to visualize γH2AX. Images were saved and transferred to NIS-Elements (Nikon) or FIJI (Schindelin et al., 2012) software for analysis.

Immunofluorescence images of cells that were previously time-lapse imaged were acquired using a Nikon Instruments Ti2-E microscope with Crest V3 Spinning Disk Confocal, Hamamatsu Orca Fusion Gen III camera, Lumencor Aura III light engine, TMC CleanBench air table, with 40× air objective (N.A 0.75, W.D. 0.66, MRH00401), and a 12-bit camera through Nikon Elements software. Images were taken at 0.5 µm z-steps over 4.5 µm. Ultraviolet light (excitation 408 nm) was used to visualize DNA via DAPI, green fluorescent light (excitation 546 nm) was used to visualize emerin or lamin B1, and red fluorescent light (excitation 638 nm) was used to visualize γH2AX, 53BP1, lamin A/C and cGAS.

### Nuclear bleb analysis

As outlined previously (Bunner et al., 2025), images were analyzed in NIS-Elements or exported to FIJI (Schindelin et al., 2012) software to analyze the intensity of each component by normalizing bleb intensity to nuclear body intensity. The body, bleb and background of each nucleus image were measured by drawing regions of interest (ROI) via the polygon selection tool. Measurements of mean intensity for each ROI were recorded and exported to Microsoft Excel. Within Excel, the background intensity was subtracted from the body and bleb. Then the average bleb intensity was divided by the average nuclear body intensity to give a relative measure, where the same average intensity would result in 1. These values were transferred to Excel and two-tailed unpaired Student's *t*-tests were performed and the data was graphed.

### DNA damage analysis

To analyze γH2AX data for the presence of DNA damage, the image background intensity was subtracted and the Cy5 channel was selected to observe γH2AX foci in the nuclei. On Nikon NIS-elements or FIJI software, binary bright spot detection was used to record γH2AX foci. A diameter of 0.5 µm was used to count foci. Data were exported to an Excel file where an average number of foci per nucleus for each treatment condition was calculated.

### Live-cell time-lapse fluorescence imaging and analysis

As previously described, we used established MEF NLS–GFP stable cell lines to quantify nuclear shape and rupture (Berg et al., 2023; Pho et al., 2024a). Images were acquired with Nikon Elements software on a Nikon

Journal of Cell Science

Instruments Ti2-E microscope, Orca Fusion Gen III camera, Lumencor Aura III light engine, TMC CleanBench air table, with a 40× air objective (N.A. 0.75, W.D. 0.66, MRH00401). Live-cell time-lapse imaging was possible using the Nikon Perfect Focus System and Okolab heat at 37°C, supplemental humidity and a 5% $CO_2$ stage top incubator (H301). Cells were imaged in single-well gridded cover glass dishes (Cellvis #D35-14-1.5GI). For time-lapse data, images were taken in 2-min intervals for 3 h over 10 fields of view. Images were captured at 50 ms exposure time, 12-bit depth and with 4% blue fluorescent light (475 nm) power. Time-lapse images were analyzed by visually tracking nuclear bleb formation and rupture throughout the movie. The frequency and recency of nuclear ruptures and bleb formation and lifetime were recorded. To determine bleb lifetime, the time of bleb formation was subtracted from the total duration of the time lapse.

LNCaP nuclei were transiently transfected with NLS–GFP via the CellLight Nucleus-GFP (Thermo Fisher Scientific, C10602) system. Cells were transfected with 50 or 100 µl of CellLight 2 days prior to imaging.

### Acknowledgements
We would like to thank HHMI, which purchased microscopes used in Bioimaging class via a grant and The Biology Department at UMass Amherst for the use of the ISB 360 facilities.

### Competing interests
The authors declare no competing or financial interests.

### Author contributions
Conceptualization: M.S.B., A.D.S.; Data curation: C.G.C., N.L., E.W., M.D.Z., G.M., K.S., L.M.C., K.E.F., N.K., S.N.F., A.P.D., E.N., J.A., A.M.C., Z.L.W., D.K.-W., L.P., H.L., M.P., K.P.; Formal analysis: C.G.C., N.L., E.W., M.D.Z., G.M., K.S., L.M.C., K.E.F., N.K., S.N.F., A.P.D., E.N., J.A., A.M.C., Z.L.W., D.K.-W., L.P., H.L., M.P., K.P.; Funding acquisition: A.D.S.; Investigation: C.G.C., N.L., E.W., M.D.Z., G.M., K.S., L.M.C., K.E.F., N.K., S.N.F., A.P.D., E.N., J.A., A.M.C., Z.L.W., D.K.-W., L.P., H.L.; Supervision: K.D., M.S.B., A.D.S.; Visualization: C.G.C., N.L.; Writing – original draft: A.D.S.; Writing – review & editing: C.G.C., N.L., E.W., K.P., M.S.B., A.D.S.

### Funding
This work was primarily supported by NIH NIGMS grant Maximizing Investigators' Research Award R35GM154928. Open Access funding provided by University of Massachusetts Amherst. Deposited in PMC for immediate release.

### Data and resource availability
All raw data is available on figshare (doi:10.6084/m9.figshare.28451933.v3). Other image data sets can be made available upon request. All other relevant data and details of resources can be found within the article and its supplementary information.

### First Person
This article has an associated First Person interview with the first authors of the paper.

### Peer review history
The peer review history is available online at https://journals.biologists.com/jcs/lookup/doi/10.1242/jcs.263945.reviewer-comments.pdf

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
