## [Peer Review File · Journal of Cell Science]

Lamin B loss in nuclear blebs is rupture dependent whereas increased DNA damage is rupture independent

Catherine G. Chu, Nick Lang, Erin Walsh, Mindy D. Zheng, Gianna Manning, Kiruba Shalin, Lyssa M. Cunha, Kate E. Faucon, Nicholas Kam, Sara N. Folan, Arav P. Desai, Emily Naughton, Jaylynn Abreu, Alexis M. Carson, Zachary L. Wald, Dasha Khvorova-Wolfson, Leena Phan, Hannah Lee, Mai Pho, Kelsey Prince, Katherine Dorfman, Michael Seifu Bahiru and Andrew D. Stephens

DOI: 10.1242/jcs.263945

Editor: Megan King

Review timeline

Original submission:	20 February 2025
Editorial decision:	19 May 2025
First revision received:	14 August 2025
Editorial decision:	10 September 2025
Second revision received:	21 September 2025
Accepted:	25 September 2025

Original submission

First decision letter

MS ID#: jcs.263945

MS TITLE: Lamin B loss in nuclear blebs is rupture dependent while increased DNA damage is rupture independent

AUTHORS: Catherine G. Chu; Nick Lang; Erin Walsh; Mindy D. Zheng; Gianna Manning; Kiruba Shalin; Lyssa M. Cunha; Kate E. Faucon; Nicholas Kam; Sara N. Folan; Arav P. Desai; Emily Naughton; Jaylynn Abreu; Alexis M. Carson; Zachary L Wald; Dasha Khvorova-Wolfson; Leena Phan; Hannah Lee; Mai Pho; Kelsey Prince; Katherine Dorfman; Michael Seifu Bahiru; Andrew D Stephens

ARTICLE TYPE: Research Article

Dear Drew,

I apologize for the delay in the review of your manuscript. We have now reached a decision on the above manuscript.

To see the reviewers' reports and a copy of this decision letter, please go to:

Reviewer 1

In this manuscript by Chu et al., the authors employed immunofluorescence and live imaging techniques to explore the behavior of Lamin B1 and Emerin at nuclear blebs and attempt to determine the contributions of nuclear blebbing and/or rupture to DNA damage. Using a combination of lamin knockdowns and DNA decompaction treatments to induce nuclear blebs in mammalian cells, they show that different cell lines (MEFs and human cancer cells) show differences between the presence of Lamin B1 and Emerin at blebs and show that these differences are rupture dependent. Additionally, they show that nuclear blebbing is associated with rupture-

independent increases in DNA damage using $\hat{I}^3\text{H2AX}$. These results confirm previous findings by others but do not provide substantive new information to the field. Still, their evidence does raise a variety of interesting questions. What factors control the propensity for blebs to rupture? What is the fate of Lamin B1 in a bleb after rupture?

Specific comments:

It has been well established that B-type lamins do not localize to NE ruptures and why that is (PMID 36301259, 35269487) something that does not come across in this manuscript.

Line 52/53: Should include additional references (PMID 36301259, 35269487, 36039758)

Line 53 should be ESCRT-III

Figure 1A maintained representative image sure looks a lot like a micronuclei. DNA level is similar to the nucleus and there appears membrane protein intensity of lamin and emerin signal suggesting intact membranes lie in between the 'bleb' and the nucleus. Other non-rupture bleb images in this manuscript are more convincing.

The figure 1 title should be "consistent loss of DNA density" instead of "loss of DNA" for clarity.

The data in Figure 1 only shows data for blebs that were present in each of the conditions, but it doesn't show how each treatment induces bleb frequency. Can the authors include a graph to show the differences in abundance of nuclear blebs in each of the treatments, similar to Figure 1 in their prior work? (PMID: 38088876).

It has been well established that B-type lamins do not localize to NE ruptures (PMID 36301259, 35269487) something that does not come across in this manuscript.

Regarding this statement: 'Our findings are also consistent with laser ablation studies where lamin B is lost upon ablation of the nuclear envelope while lamin A and lamin C remain (Kono et al., 2022).' This is technically inaccurate. All lamins seem to be lost from ruptures. It is only the BAF-dependent recruitment of A-type lamins that are soluble and nucleoplasmic that differs from the behavior of B-type or A-type lamins that are insoluble/non-nucleoplasmic due to farnesylation and do not accumulate at ruptures.

cGAS (endogenous or exogenous-tagged) is a relatively stable marker of NE rupture that could be used in fixed cells to assess the correlation with a lamin B loss at blebs and prior rupture. This would be more definitive than emerin (the accumulation of which is a decent-enough rupture reporter) and would support the GFP-NLS fixed cell immunofluorescence results.

In cells with blebs that maintain lamin B, but lose emerin, is lamin A/C also lost? It is unclear to me why A-type lamins were not evaluated here as this could have provided a much more clear sense of the mechanisms underlying nuclear blebs, why they rupture or don't rupture and what happens after they rupture. One would presume so as this is likely the cause for the loss of emerin. Are these situations effectively pre-rupture blebs (i.e. likely to rupture in the future but haven't yet). If so, are the B-type lamins somehow more dynamic than the A-type lamins and able to stretch or move into the bleb and A-type lamins cannot? Is lamin B loss in the bleb due to the rupture, or the cause of it? If the B-type lamins are present in the bleb before a rupture, but not after it, where did they go (and how)?

Figure 3 has spelling errors in the axis titles: "rupture" and "nuclear" are prominent.

Regarding the concept that DNA damage is the result of whatever leads to blebs, and not the result (or sole result) of rupture, what if the condition that leads to DNA damage is what drives the blebs, or are the blebs driving DNA damage? In other words, what comes first, the DNA damage or the bleb? Especially in light of a recent report that DNA damage can cause blebbing and ruptures through ATR-mediated phosphorylation of Lamin A/C (PMID: 37832547).

The DNA damage marker γ H2AX is not entirely reliable as it also occurs during phases of the cell cycle independent of induced DNA damage, and probably should not be the sole mechanisms to assess DNA damage (e.g. PMID: 16030261). Could it be possible that the elevated γ H2AX signal actually reflects cell cycle events and not DNA damage that correlate with blebbing, with or without ruptures. Also, could the authors comment more on the relationship of the DNA damage to the bleb in terms of location. Figure 4 suggests that the bleb is a key site of damage (and an unruptured bleb at that), but figure 3 suggests that the bleb is largely devoid of damage. The use of a live imaging with a GFP-tagged DNA repair protein (example, GFP-Ku80, GFP-53BP1) with live imaging could be an option to observe cells forming NE blebs and seeing if increases in GFP enriched puncta are observed following blebbing, with and without rupture.

Reviewer 2

SUMMARY OF THE ADVANCE MADE IN THIS PAPER AND ITS POTENTIAL SIGNIFICANCE TO THE FIELD

Summary:

This manuscript by Chu et al. investigates the molecular features and functional consequences of nuclear blebs, focusing on Lamin B1 and DNA damage. Using a combination of immunofluorescence, live-cell imaging, and diverse cell lines, the authors demonstrate that:

- * Loss of Lamin B1 in nuclear blebs is associated with nuclear rupture, as marked by Emerin enrichment and confirmed through NLS-GFP time-lapse imaging.
- * Elevated DNA damage, measured by γ H2AX foci, occurs in blebbed nuclei independently of nuclear rupture, suggesting that nuclear blebbing alone is sufficient to compromise genome integrity.

Overall, understanding the relationships between nuclear blebbing, rupture, and DNA damage is an interesting topic, and the authors made some nice observations across multiple cell lines.

SUGGESTIONS TO AUTHORS

Major Comments:

- * The authors assert throughout the manuscript including in the title that Lamin B1 loss is a consequence of nuclear rupture. While the time-lapse data (Fig. 2M) supports the idea that Lamin B1 is retained in blebs that do not rupture, the static correlation data presented in Fig. 2D-I do not robustly support a strong or consistent inverse relationship between Lamin B1 levels and Emerin enrichment, a known marker of rupture. Specifically, the scatterplots in Fig. 2D-I display considerable variability and weak correlations some with low R^2 values and no clear trend which contradicts the claim of a direct or consistent association. While it is fair to conclude that Lamin B1 is typically retained in non-ruptured blebs, the reverse (i.e., that Lamin B1 loss is a reliable marker of rupture) is not fully substantiated by the data shown. Notably, there are blebs with reduced Lamin B1 that lack emerin enrichment, and some ruptured blebs where Lamin B1 levels remain unchanged. To strengthen this claim, the authors should provide more quantitative support for instance, what proportion of Lamin B1-deficient blebs also show elevated emerin enrichment? A contingency table or summary statistics comparing these variables across cell types would provide more clarity. Alternatively, the authors should consider softening the strength of their conclusions regarding causality and revise the title and text accordingly to reflect the nuance in their findings.
- * Throughout the manuscript, the authors imply that nuclear blebbing or rupture causes increased DNA damage for example, stating that "nuclear blebbing is sufficient to induce increased DNA damage at levels comparable to those observed in ruptured nuclei" (Results, final page, lines 8-9). However, the data presented only establish a correlation between these phenomena, not a causal relationship. Crucially, the possibility that DNA damage precedes and contributes to nuclear blebbing or rupture is not addressed. Indeed, previous studies (e.g., Kovacs et al., 2023, Molecular Cell) have shown that DNA damage can itself lead to changes in nuclear integrity, including rupture. Without temporal or mechanistic evidence to support a unidirectional pathway from blebbing/rupture to DNA damage, the current data cannot distinguish cause from effect. The authors should revise the manuscript to reflect this limitation. Rather than asserting causality, it would be more appropriate to state that nuclear blebbing is associated with increased DNA damage, and to acknowledge the possibility of bidirectional or alternative relationships. If available, any temporal analysis (e.g., DNA damage markers appearing after rupture in live

imaging) could help support the causal inference, but as it stands, the conclusion should be stated more cautiously.

* While the manuscript presents important findings, its clarity particularly in the abstract and introduction would benefit from further refinement. The abstract, in particular, could be made more accessible by improving sentence flow, clarifying key results, and minimizing jargon. Additionally, the introduction would be strengthened by including a broader overview of the current state of the field. At present, the text relies heavily on self-citation, which gives a somewhat narrow view of the relevant literature. Incorporating key external studies especially those offering alternative models of nuclear rupture and DNA damage would provide valuable context and balance.

* Finally, the discussion at times blends data interpretation with speculative mechanisms. While the hypotheses offered are interesting and potentially impactful, the authors should more clearly distinguish between what is directly supported by their data and what remains hypothetical. This will help avoid overstatement and improve the overall scientific rigor of the manuscript.

Minor comment:

Typo - Page 1, line 53. ECRT is supposed to be ESCRT

First revision

Author response to reviewers' comments

In this manuscript by Chu et al., the authors employed immunofluorescence and live imaging techniques to explore the behavior of Lamin B1 and Emerin at nuclear blebs and attempt to determine the contributions of nuclear blebbing and/or rupture to DNA damage. Using a combination of lamin knockdowns and DNA decompaction treatments to induce nuclear blebs in mammalian cells, they show that different cell lines (MEFs and human cancer cells) show differences between the presence of Lamin B1 and Emerin at blebs and show that these differences are rupture dependent. Additionally, they show that nuclear blebbing is associated with rupture-independent increases in DNA damage using γ -H2AX. These results confirm previous findings by others but do not provide substantive new information to the field. Still, their evidence does raise a variety of interesting questions. What factors control the propensity for blebs to rupture? What is the fate of Lamin B1 in a bleb after rupture?

Specific comments:

It has been well established that B-type lamins do not localize to NE ruptures and why that is (PMID 36301259, 35269487) something that does not come across in this manuscript.

Line 52/53: Should include additional references (PMID 36301259, 35269487, 36039758)

Line 53 should be ESCRT-III

Figure 1A maintained representative image sure looks a lot like a micronuclei. DNA level is similar to the nucleus and there appears membrane protein intensity of lamin and emerin signal suggesting intact membranes lie in between the 'bleb' and the nucleus. Other non-rupture bleb images in this manuscript are more convincing.

The figure 1 title should be "consistent loss of DNA density" instead of "loss of DNA" for clarity.

The data in Figure 1 only shows data for blebs that were present in each of the conditions, but it doesn't show how each treatment induces bleb frequency. Can the authors include a graph to show the differences in abundance of nuclear blebs in each of the treatments, similar to Figure 1 in their prior work? (PMID: 38088876).

It has been well established that B-type lamins do not localize to NE ruptures (PMID 36301259, 35269487) something that does not come across in this manuscript.

Regarding this statement: 'Our findings are also consistent with laser ablation studies where lamin B is lost upon ablation of the nuclear envelope while lamin A and lamin C remain (Kono et al., 2022).' This is technically inaccurate. All lamins seem to be lost from ruptures. It is only the BAF-dependent recruitment of A-type lamins that are soluble and nucleoplasmic that differs from the behavior of B-type or A-type lamins that are insoluble/non-nucleoplasmic due to farnesylation and do not accumulate at ruptures.

cGAS (endogenous or exogenous-tagged) is a relatively stable marker of NE rupture that could be used in fixed cells to assess the correlation with a lamin B loss at blebs and prior rupture. This would be more definitive than emerin (the accumulation of which is a decent-enough rupture reporter) and would support the GFP-NLS fixed cell immunofluorescence results.

In cells with blebs that maintain lamin B, but lose emerin, is lamin A/C also lost? It is unclear to me why A-type lamins were not evaluated here as this could have provided a much more clear sense of the mechanisms underlying nuclear blebs, why they rupture or don't rupture and what happens after they rupture. One would presume so as this is likely the cause for the loss of emerin. Are these situations effectively pre-rupture blebs (i.e. likely to rupture in the future but haven't yet). If so, are the B-type lamins somehow more dynamic than the A-type lamins and able to stretch or move into the bleb and A-type lamins cannot? Is lamin B loss in the bleb due to the rupture, or the cause of it? If the B-type lamins are present in the bleb before a rupture, but not after it, where did they go (and how)?

Figure 3 has spelling errors in the axis titles: "rupture" and "nuclear" are prominent.

Regarding the concept that DNA damage is the result of whatever leads to blebs, and not the result (or sole result) of rupture, what if the condition that leads to DNA damage is what drives the blebs, or are the blebs driving DNA damage? In other words, what comes first, the DNA damage or the bleb? Especially in light of a recent report that DNA damage can cause blebbing and ruptures through ATR-mediated phosphorylation of Lamin A/C (PMID: 37832547).

The DNA damage marker γ H2AX is not entirely reliable as it also occurs during phases of the cell cycle independent of induced DNA damage, and probably should not be the sole mechanisms to assess DNA damage (e.g. PMID: 16030261). Could it be possible that the elevated γ H2AX signal actually reflects cell cycle events and not DNA damage that correlate with blebbing, with or without ruptures. Also, could the authors comment more on the relationship of the DNA damage to the bleb in terms of location. Figure 4 suggests that the bleb is a key site of damage (and an unruptured bleb at that), but figure 3 suggests that the bleb is largely devoid of damage. The use of a live imaging with a GFP-tagged DNA repair protein (example, GFP-Ku80, GFP-53BP1) with live imaging could be an option to observe cells forming NE blebs and seeing if increases in GFP enriched puncta are observed following blebbing, with and without rupture.

Second decision letter

MS ID#: jcs.263945R1

MS TITLE: Lamin B loss in nuclear blebs is rupture dependent while increased DNA damage is rupture independent

AUTHORS: Catherine G. Chu; Nick Lang; Erin Walsh; Mindy D. Zheng; Gianna Manning; Kiruba Shalin; Lyssa M. Cunha; Kate E. Faucon; Nicholas Kam; Sara N. Folan; Arav P. Desai; Emily Naughton; Jaylynn Abreu; Alexis M. Carson; Zachary L Wald; Dasha Khvorova-Wolfson; Leena Phan; Hannah Lee; Mai Pho; Kelsey Prince; Katherine Dorfman; Michael Seifu Bahiru; Andrew D Stephens
ARTICLE TYPE: Research Article

Dear Drew,

We have now reached a decision on the above manuscript.

To see the reviewers' reports and a copy of this decision letter, please go to:

As you will see, both reviewers support publication of your manuscript. However, I would like you to please address the suggestions that Reviewer 1 has outlined in terms of data presentation and the text as you prepare your final version of the manuscript. I hope that you will be able to carry these small changes out quickly so that I can formally accept your final manuscript.

Second revision

Author response to reviewers' comments

Comments from the Reviewers:

Reviewer 1: This revised manuscript by Chu et al., has addressed some concerns raised during the first review. There remain considerable limitations in the extent of the results that can be drawn based on the approaches utilized, namely immunolabeling of fixed cells to study dynamic events. But nonetheless this is a valuable contribution to our collective understanding of events that occur at nuclear blebs and ruptures.

We appreciate that the reviewer comments that our manuscript “is a valuable contribution to our collective understanding of events that occur at nuclear blebs and ruptures.”

Our goal is to try to bridge the divide created by the difficulty in measuring nuclear rupture via time lapse imaging on both a basic and large scale. We use time lapse immunofluorescence labeling to provide both the history of nuclear blebbing and rupture and the specific labeling of nuclear rupture-associated proteins (Figure 1). This technique is time consuming to use on multiple cell lines. Thus, we used immunofluorescence labeling across multiple cell lines using the established knowledge from time lapse immunofluorescence (Figure 2 and 3). This reveals novel data about rupture prone and averse nuclei. Using this data, we ask and answer a novel question: Is increased DNA damage associated with nuclear blebbing dependent on rupture? (Figure 4). Using the immunofluorescence labeling, which revealed cell types that might be averse to rupture, we tested one of those cell lines (LNCaP) via time lapse imaging to confirm that it in fact rarely ruptures (Figure 5). The fixed immunofluorescence data presented in Figure 2 and 3 allowed us to assay 6 cell lines across 10 conditions. This reveals these two key categories of rupture prone and averse, which we could then study further by performing time lapse imaging to confirm dynamic events. We address the reviewer's concerns point by point below.

Specific comments:

1. Figure 1C- Unclear from graph if LaB1 is in triplicate

Figure panel 1C is of biological triplicates as stated in the figure legend: “Each condition was assayed in triplicate with lamin A/C and lamin B1 n = 19,24,21, emerin n = 22, 69, 29; and cGAS n = 15, 55, 76.” We have also provided our raw data comprising this figure: <https://doi.org/10.6084/m9.figshare.28451933.v3>

As noted by Reviewer 2 “data confirming that loss of Lamin B1 occurs in 100% of observed nuclear rupture events is compelling.” We appreciate the kind words.

2. Figure 1C- Can authors correlate presence of Emerin and/or cGAS at ruptures with time from rupture (live imaging followed by fixing).

We have provided new data in the revised manuscript (Supplemental Figures 1D and 1E) to address this requested analysis. We address each individual point below.

a. These proteins should rapidly accumulate and then be lost as time increases.

We find some loss of emerin or cGAS enrichment over time. When we analyze our 3-hour time lapses of nuclear ruptures, we find a visual difference in ruptures that occurred recently (less than one hour from immunofluorescence fixation) vs. those that ruptured more than one hour before fixation. Emerin decrease in fluorescence signal in the ruptured bleb for recent vs. older ruptures is not statistically significant, while this difference is significant with cGAS. This might be due to the fact that the emerin data had a smaller sample size than that of cGAS, which would aid finding significance.

b. The suggestion that cGAS or Emerin is not accumulating at 100% of ruptures is surprising and contradictory to the literature.

Emerin is reported in Young et al., 2020 MBoC (PMID: 32459568) at ~90% of ruptures. Our recapitulates that emerin enriches to most (80%) but not 100% of ruptures.

To our knowledge there is no other publication that has measured the recruitment of cGAS and Emerin to naturally occurring bleb-based ruptures by time lapse imaging of NLS-GFP.

c. Instead, it seems more likely that these proteins are being lost from rupture sites over time, or perhaps in a few instances have just ruptured prior to fixation (although 2 minutes is more than enough time to accumulate cGAS).

As stated in response to part A of this line of questioning, we do see decreased enrichment of emerin and cGAS but significance is mixed. Nuclear blebs in MEF nuclei rupture on average once per hour, which is where the majority of the data exists (< 1 hour since NLS-GFP verified rupture). It is possible that longer term imaging (greater than 3 hours) might help determine this loss of emerin or cGAS for older ruptures. However, this line of inquiry is beyond the scope of the current manuscript.

d. Authors should do correlations time from last rupture and accumulation. This should not require any additional experiments, just further analysis of existing data. Presumably, this data would support a statement that loss of lamin B1 is a more persistent reporter of nuclear rupture compared to accumulation of cGAS or repair proteins like BAF, LEM domain proteins or A-type lamins.

We have provided the requested analysis and included it in the revised manuscript: see Supplemental Figure 1D and 1C.

e. Perhaps it would be worth discussing if B-type lamins ever recover at sites of rupture or if they can only be deposited during post-mitotic envelope reformation. We have revised the manuscript to discuss.

3.Supp Fig 2C- LaminA enrichment timing presented here doesn't fit the model/literature. Lamin A/C accumulation at ruptures is well documented, even in non-laser induced ruptures (e.g. PMID 27013428). At the very least this should be discussed. Is it possible that there are technical reasons for the discrepancy, perhaps antibodies that don't detect the phosphorylated/disassembled A-type lamins that accumulate rapidly at rupture sites?

First, we believe the reviewer mis-cited the figure. Supplemental Figure 2C is a table about emerin levels across multiple cell types. We think they meant Supplemental Figure 1B which is a scatter plot of lamin A/C bleb to body ratio relative to time since rupture.

We have revised the text to state that lamin C is recruited quickly post-rupture (Kono et al.). This same report shows that lamin A is lost post-rupture (10 min) and then recruited back (60 min). This

agrees with our data, which shows that within 60 min post-rupture there is decreased lamin A/C, then recruitment back occurs after 60 min (supplemental Figure 1B). Thus, it is likely that our lamin A/C antibody is marking largely a mix of lamin A and C behaviors.

We would like to note that the provided citation PMID 27013428 is Denias et al., which was cited in this context in the original manuscript. We believe the reviewer is referring to Figure 3I-K of the cited paper. Specifically, Figure 3K reports GFP-lamin A accumulation size in creases with NLS rupture, not a ctually overall bleb or site of rupture enrichment.

Restated, this is not a measure of intensity but instead refers to recruitment size of lamin A coming back after GFP-lamin A is lost post-rupture as shown in Figure 3J. Many labs have seen that post-rupture lamin A nucleates at the furthest point of the site of rupture and then spreads out, but this is far from enriching.

To be clear, Figure 3I, J, K shows that lamin A is lost (agreement with us and Kono et al.,) but is recruited back post-rupture. This and the following data support/agree with our findings. Figure 2B shows an example image of a nucleus with a protrusion with decreased lamin A/C stain, which agrees with loss of lamin A/C in ruptured blebs in our manuscript. Figure 3F reports 60% of cells have lamin A in the bleb, which again agrees with our heterogenous finding of lamin A/C levels. Figure 4E shows an example image of a bleb in which lamin A/C appears similar to the other parts of the lamina but is not enriched, and lamin B is absent.

Overall, we report that lamin A/C staining at the nuclear bleb can be heterogenous, which is supported by data from this paper showing that some nuclei are missing lamin A/C (Figure 2B) while some have it (Figure 4E). Overall, those that have or do not have hovers around 40-60% (Figure 3F). Supplemental Figures from this paper include many example images of nuclear blebs that rupture during confinement migration with loss of lamin A- GFP that recruits back post-rupture, but there appear to be no example images showing enrichment.

4. Fig 2 and 3 is difficult to interpret, due to a lack of a rupture reporter independent of Emerin and lamin B1. We are left assuming that blebs with a loss of lamin B1 and accumulation of Emerin have ruptured, and all other conditions are unruptured blebs. The authors seem reluctant to overtly state upfront that these two conditions initially mentioned in Fig 2 are ruptured and unruptured blebs, perhaps because there isn't an independent rupture reporter (e.g. cGAS).

Loss of lamin B1 and enrichment of emerin immunofluorescence signals as nuclear rupture markers were confirmed via time lapse NLS-GFP (Figure 1). The novelty of Figures 2 and 3 were to assay a cross multiple conditions and cell lines using these static reports of nuclear rupture. Many past publications have only tested a single cell line for nuclear rupture behavior. Our novel data, presented in Figures 2 and 3, assays 6 cell lines and 10 overall conditions, providing unique insights using these static rupture reporters. Specifically, this approach revealed that there are rupture prone and rupture averse cell lines. This experiment provides the setup for Figure 5, in which we determine if rupture averse cell lines can be confirmed via live cell NLS-GFP rupture experiments. Furthermore, this provides unique insights into the impact of nuclear rupture, or lack thereof, on increased DNA damage associated with nuclear blebs (Figures 4 and 5).

The independent rupture reporter is NLS-GFP, used in Figure 1 to lead into Figure 2 and 3, then followed up in Figure 5.

5. Fig 3. In the results section, the authors state "We hypothesized that observing nuclear rupture events should lead to a correlation of lamin B1 and emerin levels within nuclear blebs." We would assume that the authors intended to say the following: "Based on prior studies, we hypothesized that observing nuclear rupture events should lead to a negative correlation of lamin B1 and emerin levels within nuclear blebs. Or preferably, more specifically a loss of lamin B1 and accumulation of emerin." It is known that lamin B1 is missing in ruptures and Emerin accumulates at ruptures.

We have made the suggested revision to the manuscript.

6. Fig. 3C, what does the red box signify in the correlation graph?

We have updated the figure legend to describe the red box. "Red box in panel C denotes decreased lamin B1 and emerin."

7. Fig 3J- 2 decimal points in R-square value

We thank the reviewer for catching this typo. We have corrected the figure in the revised manuscript.

8. Fig 3M- Could the authors include the missing data in this graph, so 100% of the data is included, specifically LB1 maintained, Emerin maintained (presumed unruptured), and LB1 lost, Emerin enriched (presumed ruptured)

Figure 3M analyzes differences between rupture prone and averse cell types, showing clear differences using the data presented. In the original manuscript we provided the raw data (see 'Data availability') so that readers can interact with and analyze the data as they choose. We provide it here as well: <https://doi.org/10.6084/m9.figshare.28451933.v3>.

9. In the results section in the description of Fig. 3M, "However, not all lamin B1 decreased nuclear blebs showed emerin enrichment, as some measured similar or reduced emerin in the nuclear bleb relative to the body (Fig. 3M)", would this mean that loss of lamin B1 at the bleb is not a ubiquitous marker of NE rupture, since there is no evidence of Emerin increase? Or, is this evidence of a rupture that occurred a long time ago, such that Emerin levels have reduced to normal and lamin B1 has not recovered? These uncertainties are likely unavoidable when performing experiments on fixed cells with no record of nuclear rupture, but should be addressed more clearly.

Yes, we agree with the line of thinking of the reviewer. Since our time lapse to immunofluorescence experiments imaged for three hours, we may not have captured the true length since last rupture. The main take home of the data is that lamin B1 is lost upon nuclear rupture consistently while other rupture markers emerin and cGAS do not provide consistent labeling, which newly included data shows lose some enrichment over time since rupture -though significance is mixed (Supplemental Figure 1D and E). Why this occurs for emerin and cGAS remains a question beyond the scope of the current manuscript.

10. Could the authors explicitly state that the location of the DNA damage is not consistently correlated to the location of the bleb?

We have added an explicit statement to the second paragraph of the DNA damage results section in the revised manuscript.

11. The authors should mention that the loss of lamin B from the envelope, either in conjunction with cGAS enrichment or DNA protrusion from nuclear circularity is an previously reported method to identify ruptures (methods in PMID: 37832547)

We have cited this work in the revised manuscript in reference to using cGAS as a rupture marker in the introduction. We also cite this work in the discussion about the loss of lamin B1 alongside cGAS recruitment to measure nuclear ruptures.

12. The authors should ensure that all experiments list the number of experimental replicates performed. For instance, Supp Fig 3C lacks this information.

We thank the reviewer for highlighting this mistake and we have added the number of replicates to the Supplemental Figure legend.

13. In the methods, nuclear bleb analysis, the authors have written: These values were transferred to Prism where Mann-Whitney test, unpaired t-tests, or one-way ANOVA were performed, and the data was graphed; not sure this is accurate. Presumably this is a residual note from manuscript preparation and need to be removed.

We thank the reviewer for highlighting this mistake and we have removed this from the method section of the revised manuscript.

Reviewer 2: The addition of time-lapse imaging data confirming that loss of Lamin B1 occurs in 100% of observed nuclear rupture events is compelling. This new evidence convincingly demonstrates that Lamin B1 loss is a highly reliable marker of nuclear rupture, even outperforming previously established markers such as Emerin and cGAS. The new analyses (Figures 1 and 3M) significantly strengthen the conclusions and clarify the relationship between Lamin B1 and nuclear rupture. The authors have satisfactorily addressed all major and minor concerns. The revised manuscript is significantly strengthened, both in terms of data and presentation. I am satisfied with the revisions and believe that the manuscript is now ready for publication.

We thank the reviewer for their valuable feedback, all of which has greatly improved the quality of our manuscript.

Third decision letter

MS ID#: jcs.263945R2

MS Title: Lamin B loss in nuclear blebs is rupture dependent while increased DNA damage is rupture independent

Authors: Catherine G. Chu; Nick Lang; Erin Walsh; Mindy D. Zheng; Gianna Manning; Kiruba Shalin; Lyssa M. Cunha; Kate E. Faucon; Nicholas Kam; Sara N. Folan; Arav P. Desai; Emily Naughton; Jaylynn Abreu; Alexis M. Carson; Zachary L Wald; Dasha Khvorova-Wolfson; Leena Phan; Hannah Lee; Mai Pho; Kelsey Prince; Katherine Dorfman; Michael Seifu Bahiru; Andrew D Stephens
Article Type: Research Article

Dear Drew,

I am happy to tell you that your manuscript has been accepted for publication in Journal of Cell Science, pending standard publication integrity checks.